Researching COVID-19 tracing app acceptance: incorporating theory from the technological acceptance model

http://orcid.org/0000-0003-2300-6256 Velicia-Martin Felix 1
http://orcid.org/0000-0001-5723-3153 Cabrera-Sanchez Juan-Pedro 1
Gil-Cordero Eloy 1
http://orcid.org/0000-0001-9966-0698 Palos-Sanchez Pedro R. 2 ppalos@us.es
1 Department Business Administration and Marketing, University of Sevilla , Sevilla , Spain
2 Department Financial Economy and Operations Research, University of Sevilla , Sevilla , Spain
Shang Yilun
Electronic publication date: 2021 Jan 4
Publication date: 2021
Volume: 7
Electronic Location ID: e316
Received 2020 Jul 26; Accepted 2020 Oct 23
Copyright: © 2021 Velicia-Martin et al.
Copyright year: 2021
Copyright holder: Velicia-Martin et al.
License: This is an open access article distributed under the terms of the Creative Commons Attribution License, which permits unrestricted use, distribution, reproduction and adaptation in any medium and for any purpose provided that it is properly attributed. For attribution, the original author(s), title, publication source (PeerJ Computer Science) and either DOI or URL of the article must be cited.
License URL: https://creativecommons.org/licenses/by/4.0/

Keywords: Covid-19, TAM, Privacy, APP, Technology adoption

Funding: The authors received no funding for this work.

==============================
Background

The expansion of the coronavirus pandemic and the extraordinary confinement measures imposed by governments have caused an unprecedented intense and rapid contraction of the global economy. In order to revive the economy, people must be able to move safely, which means that governments must be able to quickly detect positive cases and track their potential contacts. Different alternatives have been suggested for carrying out this tracking process, one of which uses a mobile APP which has already been shown to be an effective method in some countries.

Objective

Use an extended Technology Acceptance Model (TAM) model to investigate whether citizens would be willing to accept and adopt a mobile application that indicates if they have been in contact with people infected with COVID-19. Research Methodology: A survey method was used and the information from 482 of these questionnaires was analyzed using Partial Least Squares-Structural Equation Modelling.

Results

The results show that the Intention to Use this app would be determined by the Perceived Utility of the app and that any user apprehension about possible loss of privacy would not be a significant handicap. When having to choose between health and privacy, users choose health.

Conclusions

This study shows that the extended TAM model which was used has a high explanatory power. Users believe that the APP is useful (especially users who studied in higher education), that it is easy to use, and that it is not a cause of concern for privacy. The highest acceptance of the app is found in over 35 years old’s, which is the group that is most aware of the possibility of being affected by COVID-19. The information is unbelievably valuable for developers and governments as users would be willing to use the APP.

Introduction

The new coronavirus (COVID-19) pandemic which started in December 2019 in Wuhan City (Fidan, 2020) is different to the coronavirus cases previously written about in the literature, such as Middle East Respiratory Syndrome or Severe Acute Respiratory Syndrome (Li et al., 2020). This difference is due to COVID-19 being highly infectious, which led to a rapid increase in new cases and a worldwide outbreak (Mo et al., 2020). This situation meant that strong quarantine rules were imposed in large cities, towns, and public areas around the world to prevent further spread (Rao & Vazquez, 2020). COVID-19 poses unprecedented challenges to governments and societies around the world (Chinazzi et al., 2020).

The recommendations for isolation and mobility limitation by the World Health Organization (WHO) have been an effective strategy in containing and mitigating the spread rate of infection (Brooks et al., 2020). This has allowed more time to prepare for the next phases of the pandemic and avoid the saturation of health systems.

The confinement strategy has led to almost a complete shutdown of global economic activity. This will result in an unprecedented economic crisis over the coming quarters. In January 2020, before the COVID-19 pandemic, the growth forecast for the Global Economy in 2020 stood at 2.5% (World Bank, 2020), with the International Monetary Fund being more optimistic with forecasts of 3.3% growth. After just a few days of confinement, new economic forecasts which took the pandemic into account appeared in April 2020. The World Bank forecast a 4% contraction in the Global Economy (Maliszewska, Mattoo & Van der Mensbrugghe, 2020). In the United States, the gross domestic product (GDP) decreased at an annual rate of 5.0 percent in the first quarter of 2020 (Bureau of Economic Analysis, 2020). In the same way, the forecasts for the European Union in 2020 are terrifying both for GDP growth (−7.4%), unemployment (9.0%) and gross public debt (95.1%).

Disaster risk managers require technological support to make complex decisions when faced with a crisis (Kourti et al., 2019). Different ways of monitoring positive COVID-19 cases are emerging, in order to assist national and international mobility. Governments throughout the world have proposed different ways of doing this, such as the use of “immunity passports” (Angelopoulos, Damianou & Katos, 2020), the inclusion of blockchain technology to ensure monitoring in different areas (Nguyen et al., 2020), the use of “Smart Cities” to implement artificial intelligence in order to monitor positive cases (Allam & Jones, 2020), the use of APPs to inform citizens about other positive cases in their area (Abeler et al., 2020). This last solution is the case that will be studied in this investigation. Previous research has used APP to monitor health problems, such as hypertension (Ortega et al., 2016). It was first implemented in South Korea and Israel using geolocation technology.

Recently, in Europe, the French government launched the STOP COVID app which uses Bluetooth technology (Bradford, Aboy & Liddell, 2020) instead of geolocation and is used voluntarily.

The governments of other countries, such as Singapore, also adopted a voluntary installation regime for the app and launched a strong marketing campaign. However, the app penetration rate remained below 25% two weeks after the launch (Abeler et al., 2020). In the case of Spain, the state of alarm decreed by the government in March 2020 included tracking mobile devices to monitor physical contacts, a measure which meant a reduction in the right to privacy for health reasons. The apps that are currently available or being developed can be grouped into four categories: (a) Self-diagnosis apps, (b) Chatbots or conversational assistants which will provide information to citizens, (c) Mobility apps which supply anonymous data to databases in order to track the evolution of the disease and (d) Apps such as those planned for in Pan-European Privacy-Preserving Proximity Tracing (PEPP-PT) in which the APP detects the proximity of other potentially infected citizens, while keeping the information anonymous.

This study investigates apps in category (d) above which have been recommended by the European Commission in various Decisions and Directives (Across Legal Marketing, 2020) since Apps that help give warnings are the most promising for preventing the spread of the virus.

The most relevant APPs have been compiled by Li & Guo (2020) and we must emphasize that some of them already have more than two million downloads such as the APP developed in UK (NHS APP), Australia (COVIDsafe), Israel (The Shield), Singapore (TraceTogether) or Spain (Radar COVID).

Therefore, some governments are considering using apps that monitor infected people and their environment (Altmann et al., 2020). One of the main problems of this is the personal information that the user provides, which includes age, sex, height, weight and zip code, in addition to listing any chronic health conditions, such as heart, lung or kidney diseases, diabetes and the regular use of immune suppressants (Mayor, 2020).

Because a large majority of users are willing to use this type of APP (Kaspar, 2020), this means that the factors which indicate the intention to use and adoption of this type of app by future users must be investigated. The main objective of this study is to answer the question “What are the factors that affect whether the users will be willing to use an APP that would alert them if they have been in contact with anybody infected with COVID?”.

This means that the factors which indicate the intention to use and adoption of this type of app by future users must be investigated. The main objective of this study is to answer the question: “Would users be willing to use an APP that would alert them if they have been in contact with anybody infected with COVID?”

This study is structured in the following way: after this introduction, the proposed model will be justified, and the hypotheses will be formulated. The third section will explain the data collection process, identifying how each of the variables studied was measured, as well as identifying the software used in the analysis. The data and results are analyzed in paragraph 4, where the hypotheses raised in paragraph 2 are verified. Paragraph 5 presents the conclusions drawn, together with limitations of the study and possible future research.

Literature review

COVID-19 APP

Tracking positive cases of COVID-19 using APPs is useful for citizens because users can take precautions before going out of their houses (Altmann et al., 2020). This type of APP has many different uses, one of which is controlling the epidemic without blocking the movement of citizens and the resulting economic damage (Ferretti et al., 2020). For this reason, this study analyzes the different factors which can influence the acceptance of an app by the population.

Technology acceptance model

The technological acceptance model (TAM) proposed by Davis (1985) and Davis, Bagozzi & Warshaw (1989) is used for this study as it is widely accepted by the researchers in the literature review (Szajna, 1996; Hong, Thong & Tam, 2006; King & He, 2006; Evans et al., 2014). TAM is based on two primary variables, which are (1) independent variables that include Perceived Usefulness and Perceived Ease of Use and, (2) the dependent variable Attitude Towards Use. Davis (1989) defined Perceived Usefulness as the degree to which a person believes that using a system will improve their performance, and defined Perceived Ease of Use as the degree to which a person believes that using a system is effortless. In addition, Davis, Bagozzi & Warshaw (1992) noted that the actual use of the system is determined by behavioral intention of the user, and this intention is determined jointly by the attitude of users towards the use of the system and its perceived usefulness (Liu, 2015). This study uses an extended TAM model, which is the original TAM model with added variables, such as Trust (Gefen, Karahanna & Straub, 2003; Wu & Chen, 2005), Perceived Risk (Lee, 2009), which in this case is the perceived risk of catching the COVID-19 virus, and finally, privacy concerns (Zhou, 2011; Palos-Sanchez, Hernandez-Mogollon & Campon-Cerro, 2017).

The TAM with some updates was selected because apart from being a widely contrasted model, it has also been used to evaluate the intention of use of APPs in different sectors and circumstances (Munoz-Leiva, Climent-Climent & Liébana-Cabanillas, 2017; Chen et al., 2020; Yuesti, Ayu Asri Pramesti & Verawati, 2020).

Research model and hypotheses development

Technology acceptance and behavioral intention

COVID-19 has led to an unprecedented global crisis which has had many casualties, caused economic losses and disrupted daily activities (Remuzzi & Remuzzi, 2020). This is why many governments are considering allowing economic and social movement in the next phases of the pandemic, with innovative solutions such as using APPs to provide information to the population. Despite the recognized potential of Information Technology (IT) for health issues (Mattheos et al., 2008; Buntin et al., 2011), most IT-based health systems encounter user resistance (Kamal, Shafiq & Kakria, 2020), and reduced intention to use them. Therefore, the behavioral intention of users must be investigated using a technology acceptance model to find what influences the Intention to Use this App. This study uses the Technology Acceptance Model extended with health-related variables, as used by Hu, Griffin & Bertuleit (2016).

Attitude towards using

Attitude Towards Using is defined in the TAM model as the how a user feels about using the technology being studied (Davis, Bagozzi & Warshaw, 1992). Research can be found in academic literature for APPs that supports the relationship of the variables stated above using the TAM model (Yang & Zhou, 2011) with health applications (Cho, 2016).

Thus, we can formulate the following hypothesis:

Hypothesis 1: the attitude towards using an APP positively affects behavioral intention to use it.

Perceived risk of catching COVID-19

The perceived risk of catching COVID-19 has been adapted from the research by Napper, Fisher & Reynolds (2012). In general, the importance of risk in predicting human behavior cannot be denied. Perceived risk is defined as a how uncertain a person feels when deciding whether to do something or not (Nicolaou & McKnight, 2006). Perceived risk is a crucial variable for evaluating user acceptance of an APP in this study. The literature states that perceived risk contributes to expectations of negative consequences, which has a negative effect on intention to use (Hsieh, 2016).

Thus, we can formulate the following hypothesis:

Hypothesis 2: the perceived risk of getting COVID-19 positively affects the ehavioral Initention to use an APP.

Ease of use and perceived usefulness

The first studies using TAM to investigate public health information systems and APPS, found that Ease of Use and Perceived Usefulness were the most common and significant determinants of technology acceptance (Cho et al., 2014; Whitten, Doolittle & Mackert, 2005; Olver & Selva-Nayagam, 2000). Perceived Usefulness is defined as the extent to which a person believes that using a system will help improve performance (Davis, 1989). Ease of Use is defined as the degree to which a person believes that the use of technology will require minimal effort (Elkaseh, Wong & Fung, 2016). This study expects to find that the population will accept and use this type of APP because advantages can be gained by using it. The TAM model states that the attitude towards using any technological system is determined by the two variables, Perceived Usefulness and Perceived Ease of Use of the system (Sánchez Franco, Martín Velicia & Villarejo Ramos, 2007).

Thus, we can formulate the following hypothesis:

Hypothesis 3: the perceived ease of Use of the APP positively affects attitude towards using it.

Hypothesis 4: the perceived ease of Use of the APP positively affects its perceived usefulness.

Hypothesis 5: the perceived usefulness of the APP positively affects attitude towards using it.

Hypothesis 6: the perceived usefulness of the APP positively affects the behavioral intention to use.

Privacy

Privacy has been used in various previous studies as an extension to the TAM model (Zhou, 2011; Ambrose & Basu, 2012). It can be defined for health APPs as how safe a user feels that the personal health information shared with a technological system, in this case an APP, will only be used for the stated purpose and not shared with others (Kamal, Shafiq & Kakria, 2020). In previous studies (Palos-Sanchez, Saura & Martin-Velicia, 2019), privacy has been shown to be a determining factor for the acceptance of technology (Lin & Kim, 2016). The importance of privacy, when it comes to sharing medical information, cannot be denied. Therefore, users who are concerned about the privacy of sharing their personal information with a health APP may be reluctant to adopt it.

Thus, we can formulate the following hypothesis:

Hypothesis 7: privacy concern with an APP negatively affects the behavioral intention to use it.

Trust

Previous studies have included external factors such as trust in theoretical TAM models to explain the acceptance of APPs in different areas including medical services (Munoz-Leiva, Climent-Climent & Liébana-Cabanillas, 2017; Beldad & Hegner, 2018; Suki & Suki, 2017). Trust has been defined as the faith of users/patients that the services provided by a new technology will have positive results (Kamal, Shafiq & Kakria, 2020). This study analyzes users’ trust in the use of APP technological to increase their intention to use APPs that use geolocation for the COVID-19 virus.

Thus, we can formulate the following hypothesis:

Hypothesis 8: trust in an APP positively affects the behavioral intention to use it

Figure 1 shows the proposed research model using the hypotheses given above.

Figure 1 Proposed research model.

Materials and Methods

A sample of 482 possible users of the APP was used to investigate the proposed hypotheses. The surveys were written an online form which was distributed online, by email and on social networks. Previous to this phase, an initial test survey had been carried out with researchers and potential users in order to check the survey and questions and detect any possible errors.

The demographic grouping of the sample is shown in Table 1, and an interesting point which was noted is that there were 164 possible users between 15 and 30 years, 122 between 31 and 45 years and 196 over 46 years. The original measurement scales for the variables included in TAM, proposed by Davis, Bagozzi & Warshaw (1989), were adapted for this study, as were those of Son & Kim (2008) for Privacy Concern, Pavlou & Gefen (2004) for Trust, and Napper, Fisher & Reynolds (2012) for COVID. Scales used are shown in Annex 1. All variables have been measured with a seven-point Likert scale.

Table 1 Characteristics of the sample (n = 482).

Variable	Frequency	%	
Gender			
Male	211	43.78	
Female	271	56.22	
Habitat			
Between 10.000 and 20.000 inhabitants	50	10.37	
Between 20.000 and 50.000 inhabitants	57	11.83	
Between 50.000 and 100.000 inhabitants	52	10.79	
More than 100.000 inhabitants	272	56.43	
Less than 10.000 inhabitants	51	10.58	
Job			
Stay-at-home parent	16	3.32	
Student	124	25.73	
Pensioner	27	5.60	
Unemployed	28	5.81	
Employed	246	51.04	
Self employed	41	8.51	
Marital Status			
Married	226	46.89	
Divorced	20	4.15	
Separated	4	0.83	
Single	200	41.49	
Widower	8	1.66	
Living together	24	4.98	
Education Level			
Post-Graduate	120	24.90	
Primary	10	2.07	
Secondary	154	31.95	
No schooling	1	0.21	
Higher	197	40.87	
Age			
Between 15 and 30 years	164	34.03	
Between 31 and 45 years	122	25.31	
>46 years	196	40.66	

The complete model was tested with Partial Least Squares (PLS), technique for analyzing complex relationships between latent variables that allows explaining the observed data and predictive analysis as a relevant element in scientific research, which was applied using SmartPLS-3 software (Ringle, Wende & Becker, 2015).

Results

The reliability and validity of the measurement model was tested analyzing Cronbach’s Alpha, and Composite Reliability, which meant that the latent variables needed to have a minimum value of 0.7 to be acceptable (Henseler et al., 2014). Table 2 shows that the values obtained for the measurement model meet these requirements. The reliability of the constructs can be seen to be acceptable, as the values of composite reliability and Cronbach’s Alpha pass the minimum required value of 0.7 (Nunnally, 1978) and the value for the average variance extracted (AVE) exceeds 0.5 (Straub, Boudreau & Gefen, 2004).

Table 2 Composite reliability and convergent validity.

		Items	Cronbach’s alpha	Composite reliability	Average variance extracted (AVE)	
Attitude	AU1	0.957	0.961	0.975	0.928	
AU2	0.975	
AU3	0.958	
Behavioral intention	BI1	0.962	0.964	0.977	0.933	
BI2	0.967	
BI3	0.968	
Privacy concern	PC1	0.891	0.934	0.938	0.834	
PC2	0.906	
PC3	0.937	
PC4	0.913	
Perceived ease of use	PEOU3	0.890	0.891	0.925	0.755	
PEOU4	0.916	
PEOU5	0.897	
PR COVID	PRCOV2	0.826	0.759	0.861	0.675	
PRCOV7	0.879	
PRCOV9	0.756	
Perceived usefulness	PU1	0.885	0.896	0.935	0.828	
PU2	0.914	
PU3	0.930	
Trust	TRU1	0.907	0.901	0.938	0.834	
TRU2	0.910	
TRU3	0.922	

The values shown in Table 3 were used check the discriminatory validity of the model and the constructs. The Fornell and Larcker (Ringle, Sarstedt & Straub, 2012) test was used, in which where the square root of the AVE of each latent variable was compared with that of all the other variables (Barclay, Thompson & Higgins, 1995).

Table 3 Fornell and Larcker test.

	AT	BIU	PR COVID	PEOU	PU	PC	TR	
Attitude	0.963							
Behavioral intention	0.854	0.966						
PR COVID	0.298	0.332	0.821					
Perceived ease of use	0.844	0.794	0.294	0.901				
Perceived usefulness	0.893	0.835	0.282	0.846	0.910			
Privacy concern	−0.097	−0.081	0.198	−0.078	−0.070	0.912		
Trust	0.745	0.725	0.235	0.696	0.748	−0.227	0.913	

Table 4 shows the values used to test the structural model using the path coefficients. The resampling technique called bootstrapping with 5000 test cycles was used to check which relationships were significant.

Table 4 Structural model test (path coefficients).

Hypothesis	Path (β)	t-Statistic	p-Value	
1	Attitude → behavioral intention	0.461	7.323	0.000***	
2	PR COVID → behavioral intention	0.078	3.589	0.000***	
3	Perceived ease of use → attitude	0.224	4.951	0.000***	
4	Perceived ease of use → perceived usefulness	0.846	53.406	0.000***	
5	Perceived usefulness → attitude	0.703	16.994	0.000***	
6	Perceived usefulness → behavioral intention	0.293	4.395	0.000***	
7	Privacy concern → behavioral intention	0.002	0.087	0.930	
8	Trust → behavioral intention	0.145	3.510	0.000***	
Note:

*** p < 0.001. (using a single line test and Bootstrapping with 5,000 cycles).

Some indirect effects of the relationships between the main constructs were noted and are shown in Table 5.

Table 5 Indirect effects.

	Full path (β)	T-statistic	p-Value	
Perceived ease of use → perceived usefulness → attitude	0.595	16.603	0.000	
Perceived ease of use → attitude → behavioral intention	0.103	3.924	0.000	
Perceived usefulness → attitude → behavioral intention	0.324	6.992	0.000	

The results for the model show that it has high explanatory power. This can be seen from the results shown in Table 6 for the values found for R-squared of the second-order constructs, especially Behavior Intention. In both cases of R-squared and adjusted R-squared the values are higher than the recommended value of 0.1 (Falk & Miller, 1992). According to Chin (1998), a value of 0.67 indicates high explanatory power, 0.33 indicates moderate explanatory power and 0.19 indicates weak explanatory power. The results can be seen in Table 6 and show that the value for Behavioral Intention (R-squared = 0.769) and the other variables are all much higher than the minimum level indicating high indicatory power.

Table 6 Values of R-squared for the model.

	R2	Adjusted
R2	
Attitude	0.810	0.809	
Behavioral intention	0.769	0.767	
Perceived usefulness	0.716	0.716	

The fit of the model was then checked using the values of the Standardized Root Mean Square Residual (SRMR), which had a value of 0.039 which was much less than the maximum recommended limit for a good fit of 0.08 (Henseler et al., 2014). Finally, the values for the Stone-Geisser Q2 of the model (Gefen, Rigdon & Straub, 2011) were used to evaluate its predictive power. The values obtained for Q2 were greater than 0 (Roldán & Sánchez-Franco, 2012), which shows that the model has predictive power. The values obtained are shown in Table 7

Table 7 Predictive power of latent variables.

	RMSE	MAE	Q²_predict	
Attitude	0.579	0.427	0.668	
Behavioral intention	0.606	0.440	0.636	
Perceived usefulness	0.537	0.413	0.714	

Table 8 shows the results of a multigroup analysis with PLS-MGA using permutations (Chin & Dibbern, 2010). The moderating effects of the classification and sociodemographic variables presented in Table 1 were analyzed using this multigroup analysis (Henseler, 2012) in order to test the potential moderating influence of gender, level of education and employment on the relationships in the research model. The sample was divided into groups using these variables and the measurement invariance of the composite model was found (MICOM) (Henseler, Ringle & Sarstedt, 2016).

Table 8 Multigroup analysis MGA.

Moderator/hypothesis	Path (β)	t-Statistic	p-Value	
Age	PR COVID → behavioral intention	0.092	0.023	0.046	
Privacy concern → behavioral intention	−0.114	0.979	0.041	
Education level	Perceived usefulness → behavioral intention	0.529	0.000	0.000	
Trust → behavioral intention	−0.204	0.991	0.017	
Job	Privacy concern → behavioral intention	−0.131	0.989	0.022	
Trust → behavioral intention	−0.237	0.998	0.004	

Discussion, conclusions and limitations

Discussion

Extended TAM can be seen to be a useful model for predicting intention to use a new APP that allows people who have been in contact with a positive COVID-19 case to be tracked and thus help break the chain of infection. Not only have the hypotheses of the original TAM model been met, but the model with the added variables has a higher predictive power. Hypotheses H1 Attitude → Behavioral Intention (β = 0.461; t = 7.323), H4 Perceived Ease of Use Perceived Usefulness (β = 0.846; t = 53.406) and H5 Perceived Usefulness Attitude (β = 0.703; t = 16.994) have the highest factor loading. These hypotheses are from the original TAM model, together with H3 Perceived Ease of Use Attitude (β = 0.224; t = 4.951) y H6 Perceived Usefulness Behavioral Intention (β = 0.293; t = 4.395) which are also from the original model. The analysis showed that they have a confidence level above 99.9%. This means that the theoretical TAM model can be used to investigate the adoption of apps to prevent and mitigate the effects of the COVID-19 pandemic.

The variables that contribute the most to the model are Perceived Ease of Use (β = 0.846; t = 53.406) and Perceived Usefulness (β = 0.703; t = 16.994), followed by Attitude (β = 0.461; t = 7.323). These results coincide with another recent study using the extended TAM model (Saura, Palos-Sanchez & Velicia-Martin, 2020).

The relationship for the external variable which directly influenced Behavioral Intention to Use was H8 (β = 0.461; t = 7.323) which was supported with a confidence level above 99.9%. This means that the users of the app trust it to keep its promises, meet the expectations the users have, and also take into account the interests of users as citizens and members of a community. These three items significantly condition the intention to try using the app in daily life.

The results for the external variables which were added to the original TAM model for this study of COVID PR were then analyzed. The variables were included in H2 and gave the results PR COVID Behavioral Intention to use (β = 0.078; t = 3.589). The relationship was confirmed and with a confidence level of more than 99.9%. It can therefore be seen that users’ concerns about being infected or that family members are infected with COVID-19 decisively influences the intention to use the app. Users are also concerned about becoming vulnerable to COVID-19 infection. These concerns along with the feeling of vulnerability, added to the information about the high rates of COVID-19 infection, all strongly influence the intention to try out and continue to use this app frequently in the future.

The last external variable proposed for the model Privacy Concern was not significant and only had 95% confidence level. Thus, H7 Privacy Concern → Behavioral Intention to use (β = 0.002; t = 0.087) was not significant. This means that the concern about information in the app being used inappropriately, as in other person using private information in an unforeseen manner, does not negatively influence the intention to try out and continue to use the app frequently in the future. This result has been seen in previous studies on app adoption models and can be found in the literature (Palos-Sanchez, Correia & Saura, 2019).

The extended variable that most influences the user’s decision is Trust, followed by PR COVID. In both cases, the higher the trust, or the greater the perceived risk, the greater the intention to use the APP.

The multigroup analysis showed the results did not vary for Age in H2, PR COVID Behavioral Intention (95%) and H7 Privacy Concern Behavioral Intention (95%). H2 found that the relationship is not significant in young people <35 years old. H7 showed that there was a greater concern for privacy, which had a more negative influence on young people <35 years old. However, this was not significant in either group. Level of Education was found to moderate H6 Perceived Usefulness Behavioral Intention, with 99.9% confidence level, and this relationship is only significant for users with higher education. H8 Trust Behavioral Intention (95%) is also influenced by Level of Education and the relationship is strongest for users with lower levels of education.

Type of employment moderated the relationships of H7 Privacy Concern Behavioral Intention (95%) and H8 Trust Behavioral Intention. For H7, Privacy Concern was higher in contracted and self-employed workers and housewives than in students, pensioners or the unemployed. This means that employment conditions this relationship, although the relationship was not found to be significant for any of the groups. H8 Trust Behavioral Intention (99%) was not found to be significant for the group on account of type of employment.

It is worth noting the high explanatory power of the Behavioral Intention to Use model (R2 = 0.769) and other variables such as Attitude (R2 = 0.769) and Perceived Usefulness (R2 = 0.716), as well as its predictive capacity Behavioral Intention to Use (Q2 = 0.636).

Conclusions about theory

The theoretical TAM adoption model was extended in order to improve its predictive power (Munoz-Leiva, Climent-Climent & Liébana-Cabanillas, 2017; Melas et al., 2011) and to adapt it to this new technology, as well as applying it in the exceptional case of a pandemic. All the new variables, except Privacy Concern, contributed to the model and improved the investigation results. An extended TAM model is proposed to explain the adoption of information systems and geolocation technology to combat the main means of virus propagation, which is contact between people less than two meters apart.

It was also found that the population could be persuaded to adopt this app and give up privacy, although problems could occur, not only with the loss of privacy, but also increased alerts on devices or incompatibility of the app in different countries.

The main novelty of this research is related to privacy. While other authors have focused on the privacy of the APP itself (Liu et al., 2020) or how to maintain people’s privacy (Yasaka, Lehrich & Sahyouni, 2020), our research shows that people do not care about privacy when it comes to health issues. We have also extended the TAM with a new variable for the first time that is PRCOVID or perceived risk to be infected with COVID-19. This new variable has been significant and has improve the explanation power of the TAM.

Practical conclusions

The expansion of COVID-19 has led to a pandemic with millions of infected people and thousands of deaths worldwide. The adoption of extraordinary confinement measures by many governments has slowed the rapid pace of contagion and enabled health systems to care for the sick. This confinement has caused an unprecedented contraction in the global economy. Faced with this, population and pressure groups are demanding faster de-escalation processes. In order to revive the economy, people need to move, and health authorities must be able to detect positive cases early, as well as track their potential contacts. Different methods have been proposed for this tracking, including the use of mobile apps (Li & Guo, 2020).

The economy has been gravely damaged, although the balance of health and economy must be tilted in favor of the former. Therefore, tracking technologies should be used in the de-escalation processes to increase the safety of the population and be able to isolate new positive cases and their contacts.

This type technology of has been useful in countries like South Korea, which has managed to de-escalate quickly and safely, therefore reactivating its economy thanks to apps using geolocation technology (Abeler et al., 2020; De Carli et al., 2020).

The study results show that the intention to use this app is determined by its Perceived Usefulness, especially in the case of users with higher education levels (Porter & Donthu, 2006; Kucukusta et al., 2015), and that the concern for a possible loss of Privacy is not significant (Madyatmadja, Nindito & Pristinella, 2019). When choosing between health and privacy, the user chooses health.

This information will be very valuable for app developers using geolocation and also for governments, who take the decision about its use and the loss of privacy that this implies. There have not been many cases of this loss of rights, albeit on a temporary basis, in Europe's contemporary history. Governments may find it extremely useful to know that users over the age of 35 are most concerned about infection of themselves or family members and this has a decisive influence on their intention to use the app. The feeling of being vulnerable to infection by the COVID-19 virus also influencers the user intention. Therefore, both the concern and the feeling of vulnerability added to the perception of a high chance of infection by COVID-19 exert a noticeable influence on the intention to try this app and frequently use it. These results support recent studies (Abeler et al., 2020) which showed that 75% of respondents in the UK would install the app. These authors found strong support for the use of the type of app that was investigated in this study, in four major European countries, which implies a possible pan-European app with data roaming in different countries. Decisions about launching this app must be made urgently and no time can be lost. Most importantly, the user must feel that it is a useful measure (especially for users with higher levels of education), easy to use, is respectful with information and, above all, does not cause concerns for privacy. This study therefore supports the fact that users would be willing to use this app if they trust it (especially employed users) and value its usefulness, respectful attitude and ease of use, and concern for privacy would not significantly influence the intention to use it.

Annex 1: proposed scales for the model

PRCOV1: What is your opinion of the likelihood of you or a family member becoming infected with COVID-19 (extremely unlikely, very unlikely, somewhat likely, very likely, extremely likely)?

PRCOV2: I am worried about getting infected or my family members getting infected with COVID-19 (No time, Rarely, Sometimes, A moderate amount of time, A lot of time, All the time).

PRCOV 7: I feel that we may be vulnerable to COVID-19 infection.

PRCOV 9: I think the chances of us getting infected with COVID-19 are very high: (Zero, almost zero, small, moderate, large, very large).

BI1: I intend to continue using this App in the future.

BI2: I will try to use this App in my daily life.

BI3: I intend to continue to use this App frequently.

PC1: I am concerned that information from the APP may be used inappropriately.

PC2: I am concerned that someone else may find out private information about me.

PC3: I am concerned that the information provided to the App may be used by other persons or companies.

PC4: I am concerned about providing personal information to the App service provider because it could be used in a way that is not intended.

TRU1: This App gives me confidence.

TRU2: This App will keep its promises.

TRU3: This App would take into account the interests of the citizens.

AU1: Its use would be positive for my life.

AU2: Its use would be beneficial to my family and circle of friends.

AU3: Its use would be beneficial to society.

PU1: Using this App would make me feel better about myself.

PU2: By using this App I would hope to be helping society.

PU3: The use of this App would increase my peace of mind.

PEOU1: Its purpose is clear and understandable.

PEOU2: I think that learning to use the App would be very easy for me.

PEOU3: With this App, it would be easy for me to avoid the contagion of COVID-19.

PEOU4: I would find it useful to have an App that tells me how to avoid people who have the disease.

PEOU5 The existence of this App would make it easier for me to be aware of the health problem.

Supplemental Information

Supplemental Information 1 Survey dataset.

Click here for additional data file.

Additional Information and Declarations

Competing Interests

Author Contributions

Data Availability

The authors declare that they have no competing interests.

Felix Velicia-Martin conceived and designed the experiments, performed the computation work, authored or reviewed drafts of the paper, and approved the final draft.

Juan-Pedro Cabrera-Sanchez conceived and designed the experiments, performed the experiments, performed the computation work, prepared figures and/or tables, authored or reviewed drafts of the paper, and approved the final draft.

Eloy Gil-Cordero performed the experiments, analyzed the data, performed the computation work, prepared figures and/or tables, authored or reviewed drafts of the paper, and approved the final draft.

Pedro R. Palos-Sanchez analyzed the data, performed the computation work, prepared figures and/or tables, authored or reviewed drafts of the paper, and approved the final draft.

The following information was supplied regarding data availability:

Raw data are available in the Supplemental Files.

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
