# Peer review of "Researching COVID-19 tracing app acceptance: incorporating theory from the technological acceptance model"

_PeerJ Computer Science, doi:10.7717/peerj-cs.316_

## Round 0.1 · original submission · Major Revisions

Please provide a point-to-point response to Reviewer 1 and 3 when submitting the revised version. Reviewer 2 has recommended excessive inappropriate references and he/she will not be invited for the second round review.

Reviewer 1 ·

Basic reporting

(1) Note that all acronyms/abbreviations must be defined the first time they appear in the abstract, main text, and in figures or tables. What is the TAM model? What is PLS-SEM? Unclear. Please specify.

(2) Covid-19 should be always capitalized (the lower case was spotted in some places in the manuscript).

(3) As a survey paper, appropriate references to related works are not covered sufficiently in the list, especially for the contact tracing part. A systematic review of the global deployment status of contact tracing apps is recommended as below.

J. Li and X. Guo, “Global Deployment Mappings and Challenges of Contact-tracing Apps for COVID-19,” SSRN Electronic Journal, May 2020. doi: 10.2139/ssrn.3609516 Available at https://doi.org/10.2139/ssrn.3609516

Experimental design

(1) The Materials & Methods section should be expanded to describe with sufficient information to be reproducible by other investigators.

(2) Missing clarification for this in the results section: "reliability and validity of the measurement model was tested," How did you test it?

Validity of the findings

Overall, the manuscript seems to be an application of the methods used. It adds very little to knowledge already published. The reported results and the discussions are normative and minimal, most of them have been reported/discussed online in the last few months. The authors should consider highlighting the contribution/novelty of this work.

Discussion and conclusion need to be rearranged. Most of the content in the Conclusion part should be migrated into the Discussion section.

Reviewer 2 ·

Basic reporting

* The title is not clear. I suggest changing it into "Evaluating the acceptance of COVID-19 tracing application using an extended technology acceptance model" to reflect the content of the manuscript.

* The use of the term, "COVID-19" should be consistent throughout the entire manuscript. In some parts, the authors used "Covid-19" and in others, "COVID-19".

* In the abstract, avoid using "unbelievably valuable". This would be left to be judged and perceived by the readers, not the authors of the study.

* In the introduction, before embarking into the research objective, the logical stance of the problem statement needs to be fully clarified.

* In the LR, the section "Covid-19 APP" is very short and misleading. It should be further revised to include the previous mobile Apps that have been used to curb the pandemic. Using a Table to list the previous studies would be an advantage.

* The full form of TAM is "Technology acceptance model" not "technological acceptance model". Please check through the entire manuscript.

* Under the "TAM" section, why it is important to extend the TAM with the specified variables like Trust, Perceived Risk, and privacy concerns? This is an essential point that needs to be discussed and supported by recent references.

* Under the "Research model and hypotheses development" section, the hypotheses should be placed under each factor in the model. Don't accumulate all of them at the end of the section. Some readers will be confused while reading.

Experimental design

* Under the "Materials & Methods" section, you are required to answer the following:
- What is the response rate?
- What is the sampling technique?
- There might be some bias when you collect data from different means like email and social networks. How did you handle this issue? (you might use the common method bias (CMB)).
- Why PLS-SEM is used? Why not CB-SEM?

* Remove "composite" from the caption of Table 2.

* For discriminant validity, you have relied on Fornell and Larcker test only. This criterion has been criticized in the past few years. You are required to test and report the results of HTMT.

Validity of the findings

* There should be a dedicated section for theoretical and practical implications. This could be added before the conclusion section.

* In the conclusion section, some short paragraphs need to be merged together.

* The items of your survey should be provided in the Appendix to allow further replications of similar studies in the future.

Additional comments

* Please refer to the following references in your manuscript:
- Arpaci, I., Al-Emran, M., Al-Sharafi, M. A., & Shaalan, K. (2021). A Novel Approach for Predicting the Adoption of Smartwatches Using Machine Learning Algorithms. In Recent Advances in Intelligent Systems and Smart Applications (pp. 185-195). Springer, Cham.
- Arpaci, I., Alshehabi, S., Al-Emran, M., Khasawneh, M., Mahariq, I., Abdeljawad, T., & Hassanien, A. E. (2020). Analysis of Twitter Data Using Evolutionary Clustering during the COVID-19 Pandemic. CMC-COMPUTERS MATERIALS & CONTINUA, 65(1), 193-203.
- Mezhuyev, V., Al-Emran, M., Ismail, M. A., Benedicenti, L., & Chandran, D. A. (2019). The acceptance of search-based software engineering techniques: An empirical evaluation using the technology acceptance model. IEEE Access, 7, 101073-101085.

Reviewer 3 ·

Basic reporting

In this paper, the authors use an extended TAM model to investigate whether citizens would be willing to accept and adopt a mobile application and understand the relationship of citizens’ perceived risk of getting COVID-19, trust, privacy concern, perceived ease of use and perceived usefulness of the app, attitude and behavioral Intention will be examined. This is a very interesting study with practical applications and proper statistical analysis. Overall, this is a clear, concise, and well-written manuscript. Several specific comments are as follows.

Introduction:
“This means that the factors which indicate the intention to use and adoption of this type of app by future users must be investigated. The main objective of this study is to answer the question "Would users be willing to use an APP that would alert them if they have been in contact with anybody infected with COVID?””
Such a statement may be over-generalized. Why does this study explore whether citizens would be willing to accept and adopt a mobile application that indicates if they have been in contact with people infected with COVID-19? Please provide some references.

Literature Review:
For the quality of the literature review, please provide a context for why based on the extended TAM model, citizens are willing to accept and adopt a mobile application to stay in touch with people infected with COVID-19 (including citizens' perceptions of the risks of getting COVID-19, trust, privacy concerns, ease of use and usefulness of the application, attitudes and behavioral intentions).

Research model and hypotheses development:
Please move the hypotheses H1-H8 into each section (e.g. Ease of Use and Perceived Usefulness, Trust...etc), and make the hypotheses more explicit.
In the measurement of this study, please provide the reference for the questionnaire and the reliability of each construct.

Results:
Need to report descriptive statistics for the measures; for example, have the normality assumption and the multivariate normality satisfied for the purpose of structural equation modelling in this paper?

Discussion:
Besides the descriptions of the results of the Hypotheses, more discussion should be offered in more depth.

Conclusions:
Based on these findings, have the current data (yours and others) supported or differentiated? Need to link the results to extant literature, clearly articulate the knowledge gained as results of this study and how this knowledge can be used.

Experimental design

Research model and hypotheses development:
Please move the hypotheses H1-H8 into each section (e.g. Ease of Use and Perceived Usefulness, Trust...etc), and make the hypotheses more explicit.

In the measurement of this study, please provide the reference for the questionnaire and the reliability of each construct.

Validity of the findings

Results:
Need to report descriptive statistics for the measures; for example, have the normality assumption and the multivariate normality satisfied for the purpose of structural equation modelling in this paper?

Discussion:
Besides the descriptions of the results of the Hypotheses, more discussion should be offered in more depth.

Conclusions:
Based on these findings, have the current data (yours and others) supported or differentiated? Need to link the results to extant literature, clearly articulate the knowledge gained as results of this study and how this knowledge can be used.

Additional comments

no comment

---

## Round 0.2 · Minor Revisions

Please revise the paper according to the reviewers' suggestions.

Reviewer 1 ·

Basic reporting

Further English editing and proofreading should be conducted.
Some of the grammatical errors as mentioned in the first round have yet been fixed, e.g. the "Covid-19" should be capitalized, including in the title and Figure 1.

Experimental design

Most of the concerns have been answered and corrected.

Validity of the findings

After a previous review process from which the manuscript has been revised, the present reviewer appreciates the present version and the effort that the authors put into the revision.

Reviewer 3 ·

Basic reporting

This paper provides evidence of the use of a well-known method on a meaningful set of data; both the findings and the application of the successful application of the method are worth sharing with the PeerJ Computer Science community.

Experimental design

The method is appropriate and the research question well defined, relevant & meaningful.

Validity of the findings

The Results are properly and critically described.


Please check and adjust the following values to 0.7?
"The reliability and validity of the measurement model was tested analyzing Crombach’s Alpha, Composite Reliability and Average Variance, which meant that the latent variables needed to have a minimum value of 0'7 to be acceptable [63]."

Additional comments

This revised version is much improved as the author(s) have really taken on board the reviewers' comments and suggestions. The paper is now much more meaningful and worth sharing with the PeerJ Computer Science community.

---

## Round 0.3 · accepted · Accept

All reviewers' comments have been addressed. The paper can be accepted.